# Scheduling with Complex Consumptive Resources for a Planetary Rover

**Wayne Chi, Steve Chien, Jagriti Agrawal**
Jet Propulsion Laboratory
California Institute of Technology
4800 Oak Grove Drive
Pasadena, CA 91109
{firstname.lastname}@jpl.nasa.gov

## Abstract

Generating and scheduling activities is particularly challenging when considering both consumptive resources and complex resource interactions such as time-dependent resource usage. We present three methods of determining valid temporal placement intervals for an activity in a temporally grounded plan in the presence of such constraints. We introduce the *Max Duration* and *Probe* algorithms which are sound, but incomplete, and the *Linear* algorithm which is sound and complete for linear rate resource consumption. We apply these techniques to the problem of scheduling awfor a planetary rover where the awake durations are affected by existing activities. We demonstrate how the *Probe* algorithm performs competitively with the *Linear* algorithm given an advantageous problem space and well-defined heuristics. We show that the *Probe* and *Linear* algorithms outperform the *Max Duration* algorithm empirically. We then empirically present the runtime differences between the three algorithms. The *Probe* algorithm is currently base-lined for use in the onboard scheduler for NASA's next planetary rover, the Mars 2020 rover.

## Introduction

In many space missions, consumptive resources such as energy or data volume limit the number of activities that can be scheduled. These consumptive resources are oftentimes replenished periodically or gradually over time. For example, data is downlinked—replenishing data capacity—or energy is generated by solar panels or radioisotope thermoelectric generator (RTG) power supplies. The scheduler must therefore schedule activities while staying aware of resource replenishment in order to ensure that the resource state does not violate constraints (e.g. energy below a specified level or data buffers overflow). We focus on awake and asleep scheduling for a planetary rover, but our techniques generalize scheduling in the presence of complex consumptive resource activities.

We focus on the onboard scheduler for NASA's next planetary rover, the Mars 2020 (M2020) rover (Jet Propulsion Laboratory 2018a). Since the heart of our paper is awake and asleep scheduling, we concentrate on energy as the limit-

ing consumptive resource. The M2020 rover's power source is a Multi-Mission Radioisotope Thermoelectric Generator (MMRTG) (Jet Propulsion Laboratory 2018b). The MMRTG constantly generates energy for the rover's battery, but the CPU's awake and "idle" state (i.e. no other tasks) consumes more energy than the MMRTG provides. Therefore, the rover can only increase its energy, measured as battery state of charge (SOC), when the rover is asleep. The rover, however, must stay awake to not only execute activities, but also (re)-invoke the scheduler to generate a schedule. The M2020 onboard scheduler is responsible for generating and scheduling these awake periods.

In order to generate and schedule awakes, the scheduler must compute valid start times for awakes and activities jointly to ensure that there is sufficient energy for both the awake and the activities. Each activity, however, requires varying awake sizes depending on existing awake periods and the activity's scheduled start time. If the activity is close to an existing awake, it may be necessary to extend an existing awake rather than generating a new awake as this would require the rover to shutdown and wakeup in quick succession (Figure 1) which may lead to issues if the shutdown runs longer than nominally expected. Due to its varying duration, an awake's energy consumption and valid start times are challenging to determine.

The remainder of the paper is organized as follows. First, we describe the timeline representation, which is also used by the M2020 onboard scheduler. We discuss calculating valid start time intervals—intervals in which starting the activity would not violate any constraints—and define the problem in relation to the timeline framework. Second, we discuss a general case-by-case approach to handling automatically generated awakes and the challenges specific cases pose. Third, we present three specific approaches to handling these challenges when generating and scheduling awakes: a) an over-conservative approach that always uses the maximum awake period potentially required by the activity when calculating valid intervals; b) a "probing" approach that only considers a single point in time rather than the entire interval; and c) a linear algebra approach that calculates exact valid intervals given the linear rate of energy replenishment and consumption. The "probing" approach is currently base-lined for the M2020 onboard scheduler. Fourth, we present empirical analysis to compare their de-

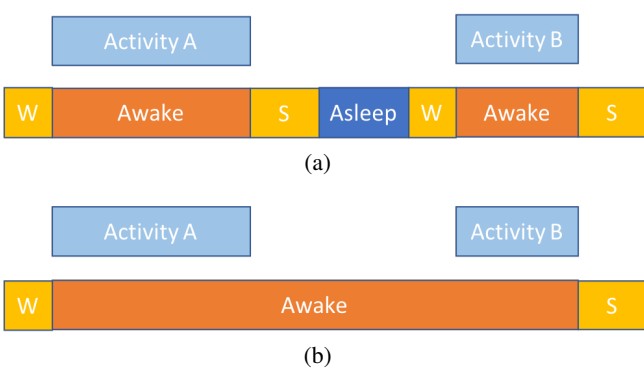

(a)

(b)

Figure 1: When scheduling activity B, the scheduler should extend the existing awake rather than creating a new one to account for the possibility that the shutdown runs longer than nominally expected. W is a wakeup and S is a shutdown.

grees of completeness and runtime performance. Lastly, we reference related works, describe future works, and discuss conclusions.

## Timeline Representation

Timelines are commonly used to model resource utilization and temporal constraints (Chien et al. 2012) and are used for the M2020 onboard scheduler as well. The timeline framework used for the M2020 onboard scheduler projects the impact of activities on shared states and resources (Rabideau and Benowitz 2017). The following is a summary of the timeline library; more detail can be found in Rabideau and Benowitz 2017.

A timeline, $T_1\langle N_1, C_1, B_1\rangle \ldots T_l\langle N_l, C_l, B_l\rangle$, is a collection of:

- Timeline impacts, $n \in N_i$, which are a change in the timeline at a specific time, $t(n)$, such as a value assignment or a change in incremental rate.

- Timeline constraints, $c \in C_i$ which are maximum or minimum limits for timeline values over a period of time; if a value exceeds the limit then there is a conflict.

- Timeline bounds, $b \in B_i$, which are maximum or minimum range for values. Values that fall outside the range are truncated, but not conflicts. This is useful to portray the maximum battery capacity, for example.

- Timeline values—where $v_i(t(u))$ is the value (e.g. SOC, current state) at time $t(u)$ for timeline $T_i$—may be calculated from the timeline impacts, $N_i$.

Each timeline also has the ability to find a) any conflicts that currently exist on the timeline and b) valid intervals for a new set of impacts, $N'$. Activities scheduled in their valid intervals create no new conflicts on the timeline. Methods (a) and (b) can also be limited to certain intervals.

There are multiple types of timelines based on practical and specific use cases (Chien et al. 2012; Rabideau et al. 1999; Knight, Rabideau, and Chien 2000), but we focus in particular on the Cumulative Rate timeline. Cumulative Rate timelines allow changes in incremental rate in addition to changes in value. This allows us to represent the incremental SOC drained and gained from being awake and asleep.

**Valid Intervals**  Typically, valid intervals for a Cumulative Rate timeline are calculated by taking in activities as a set of impacts, $N'$, and temporarily placing $N'$ at the time of each existing impact, $n \in N$. Activities are each represented as a set of start and end impacts separated by the duration of the activity. The offset between each impact in $N'$ is fixed in relation to the earliest impact, $n'_{earliest} \in N'$. $N'$ is temporarily placed at the time of each impact, $n \in N$, such that $t(n'_{earliest}) = t(n)$. This determines if any conflicts are generated and, thus, determines $N'$'s valid intervals. The impacts in $N'$ are applied based on their offset from $n'_{earliest}$. There are potentially $\mathcal{O}(N)$ impacts making this an $\mathcal{O}(N' \cdot N)$ operation. When an impact is applied, the effects to the timeline must be propagated into the future of which there are $\mathcal{O}(N)$ potential impacts. Thus, the overall runtime is $\mathcal{O}(N' \cdot N^2)$. $N >> N'$ resulting in an effective runtime of $\mathcal{O}(N^2)$. Constant time slope and intercept calculations compute any values between impacts.

These calculations are dependent on fixed impact offsets in $N'$. When impact offsets are not fixed, as is the case with non-constant awake durations, valid interval calculations are more complex. At each impact, the offsets in $N'$ are derived from a function rather than a constant. Intercept calculations further exacerbate this complication as values can change between impacts. Therefore, we must either heuristically determine fixed offsets or calculate valid intervals using a different algorithm. We present methods for both approaches.

## Problem Definition

Assume that the scheduler is given:

- a list of activities
  $A_1\langle w_1, d_1, e_1, r_1, Z_1, S_1\rangle \ldots$
  $A_\tau\langle w_\tau, d_\tau, e_\tau, r_\tau, Z_\tau, S_\tau\rangle$,

- where $w_i$ is the scheduling priority of $A_i$,

- $d_i$ is the nominal, or predicted, duration of $A_i$,

- $e_i$ is the rate at which the consumable resource energy is consumed by $A_i$,

- $r_i$ is the preferred start time for $A_i$,

- $Z_i$ is the set of physical rover zones or instruments (e.g. arm, mastcam) $z_{i_1} \ldots z_{i_k}$ that $A_i$ requires to be heated before and during use,

- $S_i$ is the set of start time windows $s_{i_1} \ldots s_{i_q}$ that $A_i$ must respect.

The scheduler is also given a global minimum SOC constraint, $C_{soc}^{min}$. Each activity may also require the automatic generation of: 1) a set of preheat activities, $P_i = \{p_{i_1} \ldots p_{i_k}\}$, 2) a set of maintenance heating activities, $M_i = \{m_{i_1} \ldots m_{i_k}\}$, or 3) an awake activity, $a_i$. Preheats are setup activities (i.e. they occur before the activity), while maintenance heating and awakes are companion activities (i.e. they occur during or with the activity).

Our goal is to calculate valid intervals for activity $A_i$ with a focus on its required awake activity, $a_i$. For this paper, we

**Algorithm 1** General Scheduling Algorithm

**Input:**
   $A\langle w, d, e, r, Z, S\rangle$: List of activities and their attributes and resources
   $C$: Constraints for the whole plan (e.g. available cumulative resources)
   $E$: Current state of the spacecraft (state of charge, data volume, activity status)
**Output:**
   $U$: Resulting schedule
1: $U \leftarrow \emptyset$
2: Sort($A$)                   ▷ By highest to lowest priority
3: **for each** $A_i \in A$ **do**
4:     $P_i \leftarrow \emptyset$
5:     $M_i \leftarrow \emptyset$
6:     **if** $Z_i \neq \emptyset$ **then**
7:         $P_i \leftarrow generate\_preheats(A_i\langle Z_i\rangle)$
8:         $M_i \leftarrow generate\_maintenances(A_i\langle Z_i, d_i\rangle)$
9:     **end if**
10:    // Consider $S_i$ as a set of disjoint valid intervals
11:    $I \leftarrow \emptyset$
12:    **for each** $S_{i_j} \in S_i$ **do**
13:        $I \leftarrow I \bigcup Split(s_{i_j})$      ▷ Based on four cases
14:    **end for**
15:    Sort($I$)                   ▷ By proximity to $t$
16:    **for each** $I_j \in I$ **do**
17:        // Calculate $I_j$ using either
18:        // Max Duration, Probe, or Linear
19:        $I_j, a_i \leftarrow valid\_intervals\_with\_awake($
                $A_i\langle d, e\rangle, I_j, P_i, M_i)$
20:        **if** $I_j \neq \emptyset$ **then**
21:            // $start$ is the scheduled start time for $A_i$
22:            $start \leftarrow schedule\_activity(A_i, I_j, U)$
23:            $schedule\_activity(a_i, start, I_j, U)$
24:            **for each** $p_{i_k} \in P_i$ **do**
25:                $schedule\_activity(p_{i_k}, start, I_j, U)$
26:            **end for**
27:            **for each** $m_{i_k} \in M_i$ **do**
28:                $schedule\_activity(m_{i_k}, start, I_j, U)$
29:            **end for**
30:        **end if**
31:    **end for**
32: **end for**

refer to valid intervals as valid *start time* intervals for activity $A_i$. An awake activity is always composed of a wakeup and shutdown. When valid interval calculations involve extending an existing awake rather than creating a new one, an existing wakeup or shutdown may be shifted to match the extended awake. Wakeups are all the same duration, as are shutdowns. If $x$ is the duration:

- The MMRTG generates $g(x)$ SOC consistently.

- The rover consumes $f(x)$ SOC when it is awake and "idle".

- Thus, when the rover is awake and "idle" the net change in SOC is $h(x) = g(x) - f(x)$.

- $g(x) \propto x$ and $f(x) \propto$ x.

- $g(x) \geq 0$ and $f(x) \leq 0$.

- $|f(x)| > |g(x)|$ as more energy is consumed when awake and idle than can be generated by the MMRTG.

- $h(x)$ is negative since $|f(x)| > |g(x)|$.

The overall scheduling algorithm is described in Algorithm 1. Scheduling an awake activity mainly involves SOC, which is represented as a Cumulative Rate Timeline. Recall that we can limit the interval considered for valid interval calculations to improve runtime; we consider $S_i$ as such limiting intervals. We assume that $S_i$ is computed or given before the problem begins. In the Mars 2020 use case, $S_i$ is actually the set of intervals after all other resources (e.g. state, dependencies) are considered. These are computed before SOC is considered due to their less significant runtime. As such, they can be generalized as $S_i$, and used to improve runtime by limiting valid interval calculation ranges.

After valid intervals are calculated, the scheduler will place the activity according to its preferred time. Each activity's preferred time, $r_i$ is a soft constraint for activity, $A_i$. The scheduler will prefer to schedule the start of the activity as close to its preferred time as possible, but is not required to schedule it at that time. Although the actual M2020 scheduler allows multiple preferred times (one for each start time window), we will assume without a loss in generality that there is only one preferred time per activity.

## Interval Cases

Valid interval calculations for non-constant duration awakes are complicated for two reasons. a) Standard valid interval calculations assume that the relative time between impacts is constant. This allows the same set of input impacts to be easily and repeatedly applied at different points on the timeline. b) Knowledge about each activity's duration is usually prior knowledge and independent from where the activity will be scheduled; this allows valid interval calculations to focus on one variable (e.g. SOC) as a function of time. Determining valid intervals when duration is dependent on scheduled time is challenging because the calculation must account for multiple variables as a function of time.

In order to schedule awakes, an activity's input intervals, $s_i$, are split into smaller intervals ($I$ in Algorithm 1). Each smaller interval matches one of the four types dependent on the activities' proximity to existing awakes and constraints. These cases are:

1. *Fully Encompassed by an Existing Awake*. If the set of activities can be scheduled entirely within an existing awake, then there is no need for a new awake activity to be generated.

2. *Disjoint from Existing Awakes*. If the set of activities can be scheduled such that any new awake is completely disjoint from an existing awake, then a new awake that encompasses all the activities must be generated and scheduled.

3. *Overlap with an Existing Awake (Straddle)*. If the set of activities overlaps with an existing awake, but is not fully

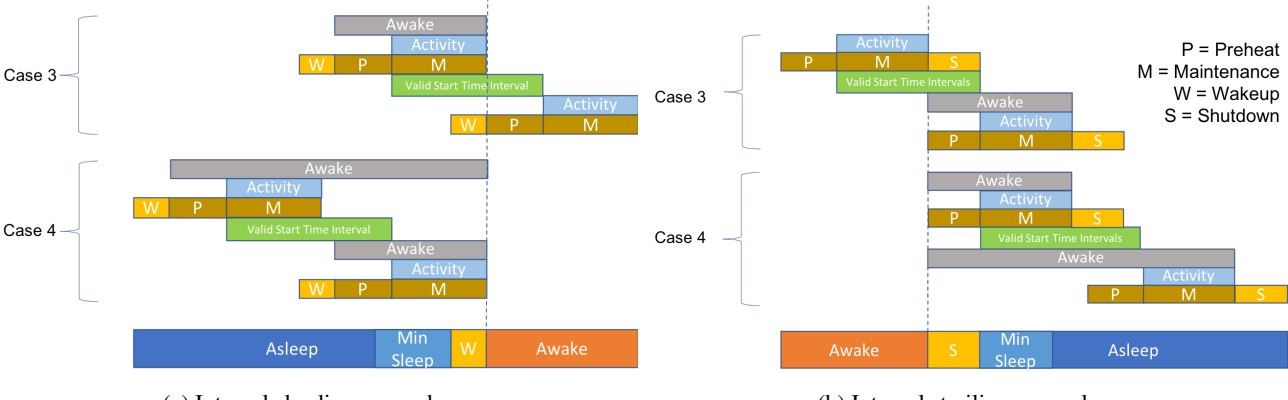

(a) Intervals leading an awake.

(b) Intervals trailing an awake

Figure 2: Intervals for each case. The awakes required at the earliest and latest times are shown. Note that these are start time intervals for activity $A_i$ given the known offset of activities in the set of activities $A_i, P_i, M_i, a_i$

encompassed by the awake, then the overlapped existing awake must be extended to encompass the set of activities.

4. *Overlap with a Minimum Asleep Constraint (Stretch).* To prevent degradation from excessive rover on-off throttling, after each shutdown the rover must stay asleep for a minimum amount of time before waking up again; therefore, there is a minimum asleep constraint both after a shutdown and before a wakeup. In addition, activities requiring an awake cannot be scheduled during a wakeup or shutdown. If the set of activities overlaps with a wakeup, shutdown, or minimum asleep constraint, then the existing awake nearest to that constraint must be extended to encompass the set of activities.

Awake duration is independent of an activity's scheduled start time for intervals matching cases 1 and 2. Case 1 requires no additional awake since it is fully encompassed by an existing awake. Case 2 requires an awake that is equal in duration to the makespan of the set of activities since there are no nearby awakes to potentially extend off of. Thus, these intervals can be handled through previously described valid interval calculations.

For intervals matching cases 3 and 4, the duration of the awake is dependent on where the activities will be scheduled. Case 3 is the *straddle* case as the activities straddle an existing awake. Case 4 is the *stretch* case as the existing awake must stretch to encompass the activities. Both the straddle and stretch cases are similar in that they require the extension of an existing awake, of which the duration (and therefore energy consumption) will vary depending on the placement of the activity. In addition, these intervals can be further categorized depending on if the extension *leads* (Figure 2a) or if it *trails* (Figure 2b) the existing awake. The scheduling algorithm splits the timeline into intervals each matching one of the above cases (Line 13 in Algorithm 1) and calculates valid intervals depending on each case (*valid_intervals_with_awake* in Algorithm 1).

In the following sections we discuss algorithms specifically designed to handle the straddle and stretch cases. Each

method describes a way to determine the awake duration. The first assumes an overestimation, the second determines exact durations, but only for a certain times, and the third computes a range of valid durations. The algorithms discussed are all sound, but some are incomplete. Violating mission constraints such as minimum battery SOC would be a significant problem (soundness), and we show in our empirical results that, for both the mission and in general, the incomplete solutions perform acceptably.

## Max Duration Algorithm

The Max Duration algorithm assumes the maximum awake duration required to schedule a set of activities. This is a simple, but over-conservative approach to handling non-constant awake duration. Let $start(I_j)$ and $end(I_j)$ be the start and end of the start time interval considered, $I_j$, for activity $A_i$. Also let $start(awake)$ and $end(awake)$ be the start and end of the nearest existing awake. The maximum awake duration is $start(awake) - start(I_j) - max(duration(p_{ik}) \in P_i)$ for leading interval cases and is $end(I_j) + d_i - end(awake)$ for trailing interval cases. Figure 2 showcases examples of the maximum awake required for both leading and trailing interval cases.

The benefit of assuming the maximum awake duration is that it allows for simpler valid interval calculations. Constant awake duration leads to a constant relative offset between impacts allowing for previously described valid interval calculations. The downside is that this approach is over-conservative. Depending on where the activities are to be scheduled, a portion of the new awake may overlap with an existing awake resulting in a "double-dipping" of resources. As the approach is over-conservative, it is sound, but incomplete; sometimes it will not find a valid interval to schedule the activities when such an interval exists.

## Probe Algorithm

The Probe approach determines the exact duration of the awake, but only for specific points of time in the input interval. Instead of computing valid intervals throughout the

entire input interval, the Probe algorithm checks for conflicts at specific points in time. At each specific point in time, the exact awake duration needed is known, thus avoiding the complications of having a non-constant awake duration.

The algorithm's simplicity is both its strength and weakness. First, the overall runtime is drastically reduced. If $k$ points in time are checked for conflicts instead of at each existing impact, then the runtime for valid intervals is $\mathcal{O}(kN)$ rather than $\mathcal{O}(N^2)$. Usually, only a few specific points are checked (e.g. earliest, latest, midpoint); hence, $k < N$. In our specific approach, we only check at the point nearest to the activity's preferred time; thus, the runtime is $\mathcal{O}(N)$. Due to this runtime improvement, the Probe algorithm is also applied to intervals of cases 1 and 2. Second, while the Max Duration algorithm is over-conservative in terms of awake duration, the Probe algorithm is exact. The downside is that the search does not span the entire interval, only "probing" certain predetermined points of time; therefore, in a sense the Probe algorithm is under-conservative in terms of the interval search space. The Probe algorithm is also sound, but incomplete. While its calculations will be accurate given its knowledge of the exact awake duration, the Probe algorithm will miss valid solutions if the probe locations are not well-defined or unlucky.

## Linear

While the other two algorithms are simple or fast, the Linear algorithm uses the linear increase in energy cost and awake duration to calculate exact valid intervals. There are two distinctions to the Linear algorithm. First, the straddle and stretch cases can be regarded as one singular *extension* case because the linear rate of energy does not change between the stretch and straddle cases. Second, the specific steps of this algorithm vary slightly depending on whether the extension *leads* the existing awake (Figure 2a) or if it *trails* the existing awake (Figure 2b); we will discuss the trailing case first.

For activity $A_i$ and input interval $I_j$ the algorithm is as follows:

1. Temporarily apply the activities to the start of the interval, $start(I_j)$, and determine if any conflicts are generated. If conflicts are generated, then there is no valid solution in $I_j$. If no conflicts are generated, then $start(I_j)$ is the start of the valid interval.

2. Temporarily apply the activities to the end of the interval, $end(I_j)$, and determine if any conflicts are generated. If no conflicts are generated, then all of $I_j$ is a valid interval. If a conflict is generated at $end(I_j)$, then a valid interval exists between $[start(I_j), end(I_j))$.

3. Recall that $N_i$ is the set of timeline impacts currently existing on the timeline $T_i$, $t(n)$ is the time of impact $n$, and $v_i(t)$ is the value at time $t$ for timeline $T_i$. Let $l$ be the point where the asleep begins between $[start(I_j), end(I_j))$. Calculate the valid interval between $[start(I_j), end(I_j))$.

    (a) Determine the point in time, $t(u)$ such that the SOC at that time, $v_{soc}(t(u))$, satisfies both a)

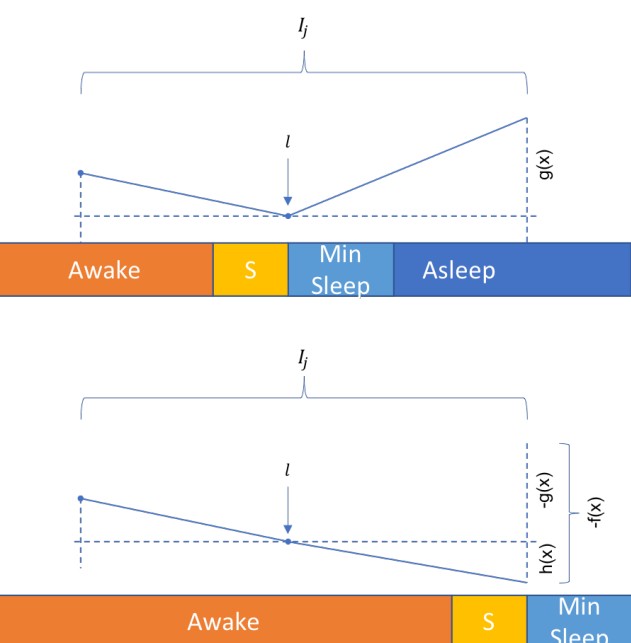

Figure 3: Energy changes from extending an awake. $l$ is the point where the asleep begins without the awake extension.

$\min_{\forall n \in N_{soc}} v_{soc}(t(n))$ and b) $t(u) > l$). In other words, $t(u)$ is the lowest point on the energy timeline after $l$.

    (b) Recall that $f(x)$ is the energy consumed while staying awake and that $C_{soc}^{min}$ is the global minimum SOC constraint. Calculate $x \ni f(x) = v_{soc}(t) - C_{soc}^{min} - energy\_cost(A_i, P_i, M_i)$.

    (c) Determine the point in time, $t(u')$ such that its SOC, $v_{soc}(t(u'))$, satisfies both a) $\min_{\forall n \in N_{soc}} v_{soc}(t(n))$ and b) $start(I_j) \le t(u') \le l$.

    (d) Recall that $h(x) = g(x) - f(x)$ is the net change in SOC while the rover is awake. Calculate $x' \ni h(x') = v_{soc}(t(u')) - C_{soc}^{min} - energy\_cost(A_i, P_i, M_i)$.

    (e) $[start(I_j), start(I_j) + min(x, x')]$ is the valid interval for the activity.

Since steps 1 and 2 are evaluated at a single point in time—similar to the Probe algorithm—the exact duration of the awake is known. For step 1, it is the minimum awake and, thus, if a conflict is generated there is no valid solution in $I_j$. For step 2, it is the maximum awake and, thus, if there is no conflict the entire range must be valid. In step 3a, we determine the maximum amount of energy that can be used without violating the minimum soc constraint as $v_{soc}(t) - C_{soc}^{min}$. In step 3b, we use this and the energy cost function to determine exactly how long the awake can potentially be. We can subtract the energy cost of the other activities $(A_i, P_i, M_i)$ because they are not affected by their placement. Steps 3c and 3d differ only slightly from steps 3a and 3b. The difference occurs because of how the awake affects the SOC timeline. In steps 3a and 3b, any

point (including the minimum point $t(u)$) after $end(I_j)$ decreases by $f(x)$ as seen in Figure 3. Recall that energy is only gained when the rover is asleep. Staying asleep between $(l, end(I_j)]$ generates $g(end(I_j) - l)$. For simplicity, let us temporarily assume $end(I_j) - l$ as $x$. Any point after $end(I_j)$ must deduct $g(x)$ since that energy will not be generated if the awake is extended. $-f(x) = h(x) - g(x)$ is the resulting energy lost from any point after $end(I_j)$. Between $[start(I_j), l]$, however, the energy has not been gained yet. As a result, $g(x)$ does not need to be deducted from any point in $[start(I_j), l]$. Using the energy function, we can calculate how long the maximum awake can be without violating the minimum SOC constraint. Step 3e simply translates the now known awake duration into valid intervals.

The main difference when applying this algorithm to the trailing case is that the existing awake extends in the opposite direction. This results in several changes. First, at $end(I_j)$ the awake duration is the minimum, and at $start(I_j)$ the awake duration is the maximum; therefore, do steps 2 and then 1. Secondly, steps 3c and 3d can be eliminated. Extending the awake from $end(I_j)$ makes it impossible for any point between $[start(I_j), end(I_j)]$ to violate the minimum SOC constraint without a point after $end(I_j)$ violating the constraint first. Lastly, in step 3e $[end(I_j) - min(x, x'), end(I_j)]$ is the valid interval.

This algorithm is both sound and complete. We can show that it is complete through a proof by contradiction. Let us assume we are scheduling in a trailing extension interval and a point, $t(u'')$, exists between $(start(I_j) + min(x, x'), end(I_j)]$ such that $A_i$ can be scheduled without violating $C_{soc}^{min}$. The awake extension duration needed is $x'' = t(u'') - start(I_j)$. We know that $x'' > min(x, x')$ otherwise it would have been in our solution. For any point after $end(I_j)$, the total energy consumed is $f(x'') + energy\_cost(A_i, P_i, M_i)$. Recall that $v_{soc}(t(u))$ is the SOC of the lowest point after $end(I_j)$. Therefore, a) $v_{soc}(t(u)) - (f(x'') + energy\_cost(A_i, P_i, M_i)) \geq C_{soc}^{min}$ otherwise the minimum SOC constraint would be violated. For any point after $end(I_j)$, the total energy consumed is $h(x'') + energy\_cost(A_i, P_i, M_i)$. Recall that $v_{soc}(t(u'))$ is the SOC of the lowest point between $[start(I_j), l]$. Therefore, b) $v_{soc}(t(u')) - (h(x'') + energy\_cost(A_i, P_i, M_i)) \geq C_{soc}^{min}$ otherwise the minimum SOC constraint would be violated. However, both statements (a) and (b) cannot be true as $f(x'') > f(min(x, x'))$. Therefore, a point between $(start(I_j) + min(x, x'), end(I_j)]$ that does not violate the minimum SOC constraint cannot exist. This proof can be similarly done for leading extension intervals.

The trade off for completeness, however, is that each valid interval requires more calculations. As a result, it increases both code complexity and runtime costs. The question is: Is the increased completeness worth the costs?

## Empirical Results

To evaluate the performance of each method, we apply each algorithm to various sets of inputs comprised of activities and their constraints. The inputs are derived from *sol types* which are currently the best available data on expected Mars 2020 rover operations (Jet Propulsion Laboratory 2018a). Each input file contains between 20 and 40 activities, and our goal is to schedule as many activities as possible. We apply our algorithms on top of the *M2020 surrogate scheduler* - a Linux workstation implementation of the same algorithm as the M2020 onboard scheduler (Rabideau and Benowitz 2017) - to construct a schedule and simulate plan execution. The M2020 surrogate scheduler is expected to produce the same schedules as the operational scheduler, but lends itself to more rapid research and development on a linux workstation environment. While the baseline scheduler utilizes the Probe algorithm, we interchange with the Linear and Max Duration algorithms; the remainder of the algorithm is identical to the operational scheduler.

We compare each method against varying incoming SOC levels to vary the level of difficulty for scheduling. The incoming SOC is the SOC remaining after the previous schedule; therefore, it is the SOC that the current schedule begins with. Since energy consumption is the main constraint and focus of sleep scheduling, varying the incoming SOC will vary the set of correct valid intervals and, by extension, the difficulty of the problem. If the incoming SOC is sufficiently high, the sleep scheduling problem is easy as there is sufficient energy for the rover to constantly stay awake and activities can be scheduled more freely. This is because each sol type was specifically constructed so that all activities would be schedulable given a reasonable (e.g. 75 percent) incoming SOC. As the incoming SOC decreases, the sleep scheduling problem gets harder as the algorithms must determine if there is sufficient energy to schedule a new awake or extend an existing awake when considering activity placement. We first analyze which algorithm performs the best as the problem difficulty increases. Secondly, we evaluate the runtimes of each algorithm.

## Results

**Completeness** Figure 4a showcases how the scheduler performs with each respective sleep scheduling algorithm. As the incoming SOC decreases, the problem difficulty increases and fewer activities are scheduled. While the conservative Max Duration approach clearly performs worse than the others, the distinction between the Probe and Linear approaches seems unclear. This may seem surprising given that the Linear algorithm is more complete than the Probe algorithm; there are, however, multiple reasons as to why the algorithms perform similarly despite their difference in completeness.

First, the scheduler focuses on finding the locally optimal state—where the local optimal is scheduling the current considered activity as close to its preferred time as possible—but does not focus on global optimality. Due to the rover's computational limitations, the M2020 onboard scheduler is a one-shot, non-backtracking scheduler. When considering each activity, the scheduler attempts to schedule it as close to the activity's preferred time as possible, but does not consider the impact it may have to any future activities. Since activities are not moved or removed after they have been considered for the schedule, the current scheduler cannot guarantee global optimality, regardless of the valid

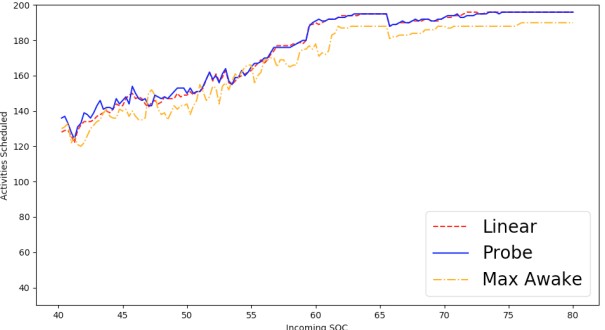
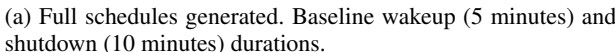

(a) Full schedules generated. Baseline wakeup (5 minutes) and shutdown (10 minutes) durations.

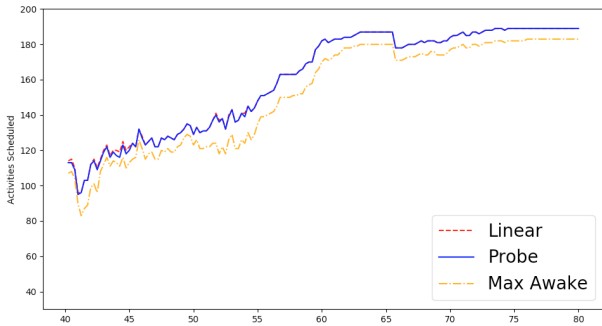

(b) Partial schedules generated. Baseline wakeup (5 minutes) and shutdown (10 minutes) durations.

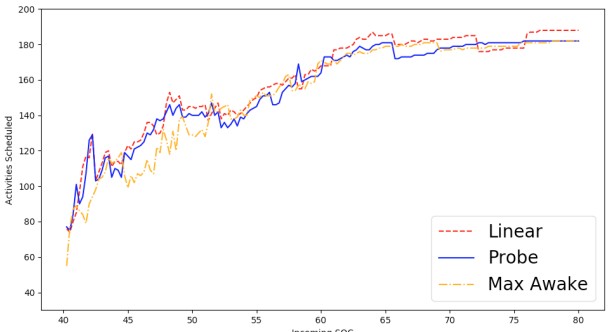

(c) Full schedules generated. Extended wakeup (30 minutes) and shutdown (60 minutes) durations.

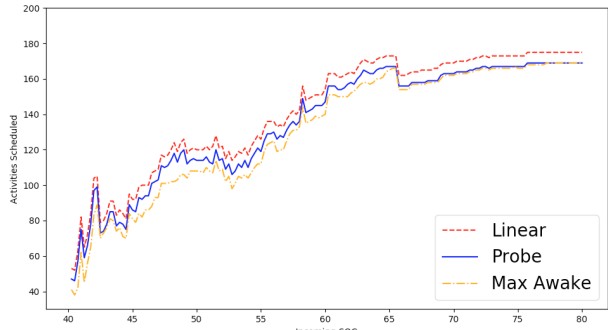

(d) Partial schedules generated. Extended wakeup (30 minutes) and shutdown (60 minutes) durations.

Figure 4: As the Incoming SOC Increases, resources are less constrained and more activities are able to be scheduled. When generating a schedule for all activities from an empty schedule, the differentiation between the Probe and Linear algorithms is unclear. When using the baseline wakeup and shutdown durations, it is clear that the Max Duration algorithm performs the worst; if the wakeup and shutdown durations are extended, however, the Max Duration algorithm starts to perform better. When scheduling only one activity at a time from a partial schedule, the Linear algorithm strictly outperforms the other two algorithms. When the wakeup and shutdown durations are extended, the difference becomes even more clear.

interval algorithm chosen. To account for this, the M2020 team has developed Copilot, a ground based tool that uses Monte Carlo and Squeaky Wheel Optimization (Joslin and Clements 1999) to adjust scheduling priorities before the plan is uplinked to the rover. To read more about how activity priorities are set, see (Chi et al. 2019).

Take for instance the following example: the Linear algorithm successfully finds valid intervals for a longer activity for which the Probe algorithm cannot. This may, however, result in the Linear algorithm being unable to schedule either multiple shorter activities or an activity that multiple future activities depend on, while the Probe algorithm (due to having excess energy from not scheduling the activity) may successfully schedule multiple future activities.

To balance the scheduler's focus on local optimality, we considered scenarios where only the next activity in the scheduling algorithm is considered. To do this, we generated partial schedules where the first $i$ activities are scheduled by the same algorithm, but the $i + 1$ activity is scheduled with the different algorithms and compared. This is repeated for every $i \in m$ so that every activity is scheduled with the same algorithm for the first $i$ activities. We used the Probe

algorithm to schedule the first $i$ activities as it is the current baseline for the M2020 scheduler. In Figure 4b, we see that the Linear algorithm strictly outperforms (although the outperformance is minimal) the Probe method while the Max Duration method strictly under-performs. These results reaffirm our initial belief that the Linear algorithm is complete and more locally optimal than the other methods. The fix to the local vs global optimality issue thus lies more with the overall scheduling algorithm than with the valid interval calculations.

Second, the stretch and straddle regions are not large enough to cause a drastic difference between algorithms. Currently, wakeups are 5 minutes, shutdowns are 10 minutes, and the minimum sleep duration is 20 minutes. The makespan of meaningful activities in a sol is around 8 to 10 hours of which there are usually only a few wakeups and shutdowns. Since the algorithms only differ in how the stretch and straddle regions—regions encompassing wakeup and shutdowns—are handled, the differences in the algorithms are minute. By increasing the length of the stretch and straddle regions, the differences between the algorithms become more clear. In Figures 4c (full schedule) and 4d (partial

schedule), we increased the wakeup duration to 30 minutes and shutdown durations to 60 minutes. Indeed, the Linear algorithm starts to perform better in comparison to the Probe algorithm with the differences especially evident in Figure 4d. The Max Duration algorithm also performs better than before as its advantage over the Probe algorithm (it searches the entire range rather than just one point) is able to be more utilized.

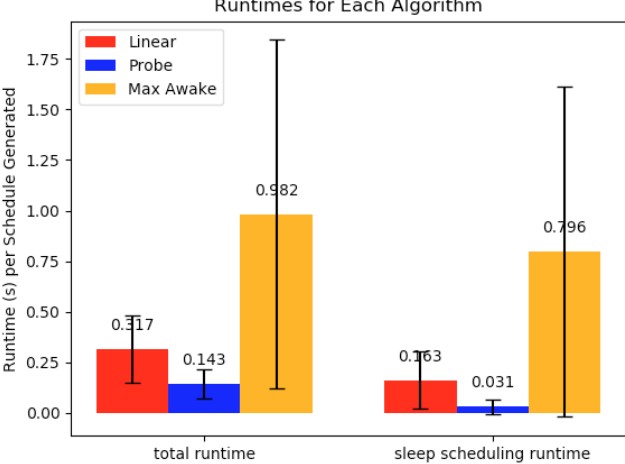

Figure 5: Left is the runtime for the overall scheduling algorithm. Right is the runtime sleep scheduling part of the algorithm only (i.e. *valid_intervals_with_awake*). The Probe algorithm greatly outperforms the other algorithms while the Max Duration algorithm greatly underperforms.

**Runtime** While completeness is important, the runtime performance of each algorithm is equally critical to determining which algorithm to use. To accurately compare the algorithms, we compare both the total scheduling runtime (left) and the runtime of the algorithms after prior computations and constant factors are deducted (right). In figure 5, the Probe algorithm significantly outperforms the other two algorithms as expected. Recall that the Probe algorithm is an $\mathcal{O}(N)$ algorithm while both Linear and Max Duration are $\mathcal{O}(N^2)$. Not only that, but the Probe algorithm can be applied to non stretch and straddle intervals as well, further speeding up runtime. Despite its simple nature, the Max Duration algorithm is the slowest algorithm by a wide margin. Since the Max Duration algorithm is often unable to find valid intervals due to its conservatism, it must often search through and calculate valid intervals for multiple stretch and straddle regions. As a result, its simple valid interval calculations is offset by its need to do this calculation multiple times for each activity. On the other hand, the Linear algorithm may take longer per region, but needs to explore less regions overall. This is further evidenced by the large standard deviation seen in the Max Duration algorithm's runtime; if it can find a valid interval within the first few regions it considers it is fast, but if it cannot then it is incredibly slow. Although the difference between tenths of a second may seem small, these computations are done on a linux workstation instead

of onboard the rover; when eventually run onboard, a single scheduler run can take up to 1 minute. As a result, the two factor increase in total runtime between the Probe and Linear algorithms is substantial.

## Future Work

There are a few topics to be considered for future research. First, preheat and maintenance heating pose a similar energy management challenge to the scheduler. While they have been intentionally glossed over for this paper, preheat and maintenance heating are also dependent on existing preheats and maintenances; an existing maintenance may be extended instead of requiring a new preheat. Preheats differ in that an activity may require preheats for multiple regions on the rover and preheat durations may vary depending on thermal conditions (Rabideau and Benowitz 2017). We would like to analyze the different algorithms used to schedule preheats and maintenances as well. Second, we would like to analyze runtimes onboard the rover. In this paper, we analyzed the runtimes on a linux workstation and compared it against an estimate of how long a scheduler run takes on a flight-like processor. The onboard scheduler runtime estimate is, however, only run with the baseline algorithm (i.e. Probe); our analysis would be further substantiated if we were able to run each algorithm onboard a flight-like processor.

## Related Work

Schedulers have a long history of handling consumptive resources.

ASPEN-EO-1 not only took into account onboard data storage (among other constraints) when scheduling plan observations, but summarized or deleted data depending on onboard data analysis (Chien et al. 2005a; 2005b; 2010).

MEXAR2 addressed Mars Express's Spacecraft Memory Dumping Problem (MEX-MDP) and synthesized data downlink plans that took into account data storage capacity (Cesta et al. 2007).

MAPGEN was used to plan operations for Mars Exploration Rovers (MER) Spirit and Opportunity(Bresina et al. 2005). MAPGEN similarly managed battery SOC as a consumptive resource, but MER rovers relied on solar power rather than a MMRTG. In addition, this system addresses ground-based rather than onboard planning.

Remote Agent is an onboard autonomous agent architecture that takes into account consumptive resources such as energy and data volume (Muscettola et al. 1998). It mainly utilizes constraint posting and propagation rather than "fixed" start times and is a more general architecture compared to our specific solution to scheduling consumptive resources.

## Conclusion

Generating and scheduling activities in the presence of consumptive regenerative resources is especially challenging when a driving factor of feasibility of placement is dependent on interactions with the existing schedule. Scheduling activities and their awake periods is particularly challenging in the context of M2020 because the awake's

duration is dependent on existing awakes. We presented three algorithms—Max Duration, Probe, and Linear—for scheduling awakes and analyzed their completeness and runtime. Despite being a locally sound and complete algorithm, the Linear algorithm was not always able to outperform in the global problem space. We demonstrated how a simple and incomplete algorithm can perform both suboptimally, as seen with the Max Duration algorithm, and also close to optimal, as seen with the Probe algorithm, dependent on the heuristic and input parameters. We showed that the Probe algorithm is a fair alternative to a more complete algorithm, especially considering its ease of implementation and runtime improvement.

## Acknowledgments

This work was performed at the Jet Propulsion Laboratory, California Institute of Technology, under a contract with the National Aeronautics and Space Administration.

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
