# OpenReview forum: "Scheduling with Complex Consumptive Resources for a Planetary Rover"
_icaps-conference.org/ICAPS/2019/Workshop/SPARK — SPARK 2019_

### Official Review · AnonReviewer2 · 2019-05-01
**Good exanple of an applications paper**

**Rating:** 5
**Confidence:** 2

**Review:**

This paper describes three alternate approaches for scheduling awake and sleep cycles for a space rover.  While awake the rover uses more energy than it produces and can only recharge its resources when it is asleep.  The three approaches vary in terms of soundness and completeness.   All approaches are sound but only one is complete.  After presenting the approaches an empirical evaluation is given comparing both the quality and the runtime of the solutions.  The evaluation shows that the naïve max duration approach. which is sound but not complete,  performs worst in terms of solution quality and runtime.  While the sound and complete linear approach produces the best quality solutions its runtime is worse than probe approach which has comparable quality.

Planning activities for a Mars rover is highly relevant to SPARK.

The initial figure could explain why it might be useful to extend an awake period while the rover is idle.  The figure clearly shows the two cases but gives no intuition as to why one case might be preferable to the other.

The probe approach depends critically on the heuristic choice of the probe point.  More discussion and/or an evaluation of strategies for choosing the probe point would be valuable given that the analysis points to the probe algorithm being a good compromise between quality and runtime,

---

### Official Review · AnonReviewer1 · 2019-05-06
**The paper addresses an interesting topic but the there are issues in the presentation and justification of proposed approaches.**

**Rating:** 3
**Confidence:** 3

**Review:**

The paper presents 3 approaches for on-board scheduling of activities in a planetary rover under reservoir resource constraints.

Two of them, the Max Duration and Probe algorithms, are sound, but incomplete, while the third, the Linear algorithm, is sound and complete for linear rate resource consumption (but more computationally expensive).

An empirical evaluation shows that the Probe algorithm (the one currently baselined for use in the onboard scheduler for NASA’s next planetary rover, the Mars 2020 rover) performs competitively and compares the runtime differences between the three algorithms.

The topic is interesting, definitely up-to-date given the current efforts in planning and scheduling for the Mars 2020 mission, and it fits the SPARK workshop.

Weakness of this paper in my opinion are in the presentation and in the justification given for the proposed approaches.

Regarding the presentation, I found the paper not easy to follow: some definitions on Timeline Representations (like the "impact" for instance) are not standard in timeline based planning (meant as the systems like EUROPA, ASPEN, APSI and others surveyed in Chien et al, SpaceOps 2012) and would have maybe deserved better explanation besides the citation to Rabideau and Benowitz 2017. Also the Problem Definition presents some issues: what are exactly the "zones" for instance?

Moreover, I would suggest to add an example, or figure, about the Probe Algorithm (at the end, since this is the chosen baseline, I think it would deserve better presentation).

On the empirical evaluation, Figure 4b, it is said that "In Figure 4b, we see that the Linear algorithm strictly outperforms the Probe method while the Max Duration method strictly under-performs". It does not look like honestly that Linear strictly outperform Probe, they look like almost the same...

Please check also the bibliography style, many citation appears incomplete.

About the justification, it is understood that the system has to operate on-board under strict computational constraints (and in this case the Probe algorithm makes sense), but some words to justify such an ad-hoc approach would have been probably appropriate. This looks like an integrated P&S problem, broken into a sequence of steps aimed at satisfying a sub-set of the problem constraints. How do you control the impact of this specific step (the calculation of awaken states) on the whole plan? There is no side-effect that could violate other logical or resource constraints?

To conclude a curiosity: in alternative to the Linear Algorithm, would have been computationally too expensive to try a proper calculation of the resource envelope (min and max subject to flexible allocation of consumptions) and then calculating a minimal/maximal flexible duration for the Asleep activities with a Max-Flow algorithm for instance?

---

### Decision · Program_Chairs · 2019-05-08
**Acceptance Decision**

**Decision:**

Accept

**Comment:**

Real-world application with NASA's Mars 2020 rover. Paper itself could be improved.